# Characterization of Leg Push Forces and Their Relationship to Velocity in On-Water Sprint Kayaking

**DOI:** 10.3390/s21206790

**Published:** 2021-10-13

**Authors:** Kent K. Klitgaard, Hans Rosdahl, Rene B. K. Brund, John Hansen, Mark de Zee

**Affiliations:** 1Sport Sciences, Department of Health Science and Technology, Aalborg University, 9220 Aalborg, Denmark; rkb@hst.aau.dk (R.B.K.B.); mdz@hst.aau.dk (M.d.Z.); 2Department of Physiology, Nutrition and Biomechanics, The Swedish School of Sport and Health Sciences, GIH, 114 33 Stockholm, Sweden; Hans.Rosdahl@gih.se; 3CardioTech Research Group, Department of Health Science and Technology, Aalborg University, 9220 Aalborg, Denmark; joh@hst.com

**Keywords:** leg forces, sprint kayak, footrest, biomechanics, force measurements

## Abstract

The purpose of this work was to describe the leg-muscle-generated push force characteristics in sprint kayak paddlers for females and males on water. Additionally, the relationship between leg pushing force characteristics and velocity was investigated. Twenty-eight paddlers participated in the study. The participants had five minutes of self-chosen warm-up and were asked to paddle at three different velocities, including maximal effort. Left- and right-side leg extension force were collected together with velocity. Linear regression analyses were performed with leg extension force characteristics as independent variables and velocity as the dependent variable. A second linear regression analysis investigated the effect of paddling velocity on different leg extension force characteristics with an explanatory model. The results showed that the leg pushing force elicits a sinus-like pattern, increasing and decreasing throughout the stroke cycle. Impulse over 10 s showed the highest correlation to maximum velocity (r = 0.827, *p* < 0.01), while a strong co-correlation was observed between the impulse per stroke cycle and mean force (r = 0.910, *p* < 0.01). The explanatory model results revealed that an increase in paddling velocity is, among other factors, driven by increased leg force. Maximal velocity could predict 68% of the paddlers’ velocity within 1 km/h with peak leg force, impulse over 10 s, and stroke rate (*p*-value < 0.001, adjusted R-squared = 0.8). Sprint kayak paddlers elicit a strong positive relationship between leg pushing forces and velocity. The results confirm that sprint kayakers’ cyclic leg movement is a key part of the kayaking technique.

## 1. Introduction

In sprint kayaking, performance relates to the paddler’s ability to propel the kayak forward by overcoming the aerodynamic drag acting on the paddler and kayak and hydrodynamic drag acting on the kayak, which is performed with a cyclic paddling movement. The cyclic paddling technique is the main contributor to the efficiency of the paddler [1,2]. The forces produced on the paddle by the paddler must be transferred through the body to the kayak. The kayaker pushes with the stroke-side leg and pulls with the contralateral leg. This leg movement rotates the pelvis towards the stroke side, which creates movement around the paddler, including an activation of the trunk muscles [3,4,5,6]. Several studies have investigated the forces produced by the paddler within the kayak. Most of these studies have been performed on kayak ergometers. Tornberg et al. [7] investigated leg forces on a kayak ergometer with a specially developed footrest, finding that the peak pushing forces on the footrest were on average 650 N for an international elite paddler. Begon et al. [7] measured forces applied on the seat and footrest on a sliding ergometer with a sliding footrest-seat complex. The seat showed predominately backwards pushing forces of 351 N ± 100 N during the draw phase of a stroke cycle with force peak pushing leg forces of 895 N.

Few studies have investigated on-water paddling kinetics [8]. Gomes et al. [9] developed a system called the ‘F-paddle system’ to evaluate the paddle’s on-water forces. This system was used by Gomes et al. [10] to quantify force profiles at different stroke rates. They found the average of the peak forces on the paddle of ≈250 N for men and ≈150 N for women. A clear difference in the paddle force profile was observed when the stroke rate changed. A study by Kong et al. [11] used the commercially available paddle system “One Giant Leap” to investigate paddle force profiles of the front and back paddlers in a large sample of K2 sprint kayaking athletes. Kong and colleagues found for the national paddlers, a paddle peak force of 344 N and 342 N for the front and back paddlers, and a paddle mean force of 205 N and 200 N for the front and back paddlers. These forces were slightly higher than the values obtained by Gomes et al. [10]. However, it is essential to note that the study by Kong et al. [11] used crew boats, where the total mass is higher than in K1. Additionally, Kong and colleagues concluded that with instrumented paddles it is possible to obtain performance feedback during on-water kayaking.

Nilsson and Rosdahl [12] also investigated on-water paddling kinetics. An on-water system was developed that measures forces applied by the paddler on the footrest and seat with individually integrated load cells [12]. This setup was used by Nilsson and Rosdahl [13] to investigate restricted versus unrestricted legs during maximal effort in sprint kayaking. The results revealed that unrestricted legs elicited higher mean and peak forces and they reported that the paddler applies pull leg forces with a mean of 220 N and peak pushing leg forces with a mean of 400 N on the footrest. Furthermore, the maximal velocity dropped by 16% when the legs were restricted compared to unrestricted. Additionally, the paddle stroke forces were registered and stroke mean forces of about 300 N with stroke peak forces of 450 N were reported [13].

The primary focus of these on-water studies has been on procedures and methods [12,14,15,16,17,18], except the studies by Nilsson and Rosdahl [13] and Gomes et al. [10], where the aim was to collect on-water data to study force profiles in the footrest and paddle [10,13]. Aside from the two mentioned studies, the role of on-water kinetics has not been extensively investigated. Inside the kayak, the force is transferred through the footrest, seat, and foot strap while the cyclic leg movement affects the forces on all three of these points. The legs push on the footrest and pull in the foot strap, which is essentially a part of the footrest. Lastly, the pull and push forces on the footrest create reaction forces in the seat [5,19,20].

The kayak footrest can be viewed as a plate, where the force acts perpendicularly to the force plate surface. Shear forces are also present; however, they are marginal compared to the perpendicular force. Only Nilsson and Rosdahl [13] have investigated leg forces during kayaking on water and we know that these reaction forces are essential to moving the kayak forwards. They concluded that the leg movement and associated leg forces are crucial for creating velocity in the kayak. It should be noted that they only investigated a small sample size (*n* = 5). Therefore, little is known about the leg force profiles’ relation to kayak velocity on the water. Measurement of leg pushing forces in the footrest concerning velocity increases would improve the understanding of how leg forces contribute to performance on the water.

This study aims to describe leg pushing force characteristics in sprint kayak paddlers for a greater sample size of mixed sex sprint kayak paddlers on water. Furthermore, we investigated the relationship between leg pushing force characteristics and velocity.

## 2. Materials and Methods

### 2.1. Participants

Twenty-eight paddlers participated in the study, from four different age groups: girls under 16 years of age, boys under 18, and senior women and men. Participant characteristics can be seen in Table 1. All participants were competitors at national or international elite level in their respective age groups. The senior men and senior women were at the highest level and included medalists in the European and World championships in both sprints and marathons. All participants were informed of the study’s nature and provided written consent to participate before each test. The study complied with the local ethical committee guidelines (Science Ethical Committee of Region Northern Jutland) and was approved.

### 2.2. Equipment

All participants used their own kayak and paddle. The leg pushing forces were measured using a custom-made footrest previously described in detail by Klitgaard et al. [21]. The footrest desing corresponds to a Nelo footrest. Therefore, it can be mounted into a Nelo kayak, as this is the most common brand on the market. The footrest consists of a metal sandwich with two load cells in between, measuring forces from the left and right leg. The footrest can be seen mounted in the kayak in Figure 1.

Data from the footrest were collected continuously at a sampling rate of 1000 Hz. The footrest consisted of two single-point load cells (LCM200 from Futek, Irvine, CA, USA), one on each side, and the footrest was split into two parts to avoid crosstalk between the right and left sides. The signals were pre-amplified using a force amplifier (Biovision, Wehrheim, Germany). With a gain factor of 10,000, set to differential recording. A portable data acquisition system acquired the data. The system consisted of a Latte Panda Windows 10 Mini PC (DFRobot, Shanghai, China) with MATLAB 2018a (Mathworks, Natick, MA, USA) installed [21]. 

Participants’ were allowed to strap their feet to the footrest with the foot strap, as they normally do in a kayak. A GPS watch (Garmin 910XT, Olathe, KS, USA) was mounted on the participants’ kayaks to monitor the velocity. The researchers followed the paddler on the water in a motorboat.

### 2.3. Protocol

The data collection was conducted during the summer under stable weather conditions for all paddlers. Before each test, the footrest was mounted to the participants’ kayaks, and the distance between the seat and footrest was the same as for the participants’ own footrests. The participants went on the water, had 5 min of self-chosen warm-up, and were asked to paddle at three different intensities, at 12 km/h for 60 s, 15 km/h for 30 s, and at a maximal effort for 20 s. The participants had rolling starts on every bout. The total workflow can be seen in Figure 2.

### 2.4. Data Analysis and Statistics

The following section has been split into a descriptive and an explanatory part. The descriptive part provides an overall understanding of leg pushing forces with force profiles and descriptive data tables, while the explanatory part investigates the more complex aspects of leg pushing forces concerning performance (maximal velocity) using statistical models.

#### 2.4.1. Descriptive Analysis

MATLAB was used for data analysis. The data were filtered using a fourth-order low-pass filter with a cutoff frequency of 20 Hz. The velocity data from the GPS watch were exported to MATLAB for analysis.

The following variables were chosen for analysis: velocity, mean leg pushing force, peak leg pushing force, stroke rate, force impulse during one stroke cycle, and force impulse over 10 s. The stroke rate was calculated from the force data by counting peaks in the timeframe. The mean push force was the average force across all participants, and the peak force was the average peak force across all participants. The force impulse values per stroke cycle and over 10 s were calculated using a trapezoidal numerical integration for one stroke cycle and over 10 s. The left and right sides were added to obtain the total impulse. Descriptive test data and participant characteristics were presented as the mean and standard deviations, although the baseline characteristics were not normally distributed. The footrest force data were averaged over 15 stroke cycles and were time-normalized to stroke cycles starting on the left side. Data were also normalized according to the maximum peaks in the footrest data to obtain the stroke cycle in percentage.

#### 2.4.2. Explanatory Analysis

Two multiple linear regression analyses were performed: one to build an algorithm to predict the maximal paddling velocity and one to investigate the effect of paddling velocity on different leg forces with an explanatory model [22]. The predictive model was checked for multicollinearity using a Pearson correlation matrix [23]. The model was fitted using the Bayesian information criterion, providing the best physically performing algorithm [24]. Each algorithm’s predictive performance was tested using a leave-one-out five-fold cross-validation approach, providing the standard deviation of the prediction in kilometers per hour. In this validation, the sample was cut into five equally sized groups, and individual kayak paddling velocities were predicted using the regression model with weights based on the data from the other four groups to which the individual paddlers do not belong [24].

The explanatory model was fitted using the different leg force characteristics as dependent variables and velocity as the independent variable. We applied a mutually adjusted model to assess potential differences in the association to account for different possible associations between velocity and leg forces across paddling levels. Both models had various aspects of the residuals evaluated, such as a residual versus fitted plot, Q–Q plot, and Cook’s distance. Moreover, the explanatory model included three observations from each participant, one for each test velocity, which may introduce a correlated design. Residuals were plotted against participants, owing to the potential for correlated observations, to assess their influence. Due to concerns about the data residuals, sensitivity analyses were performed using a bootstrap method with 1000 replications to confirm the confidence interval range. The results were evaluated for importance based on the estimates, confidence intervals, and *p*-values [25,26]. The statistical analyses were conducted in R-studio (v. 1.4.1717), except for bootstrapping, which was conducted in Stata (v. 15; Stata Corp LP, College Station, TX, USA).

## 3. Results

The results section is split into one descriptive and one explanatory parts. The descriptive part provides an overall view of the findings without a deeper analysis, providing a greater general understanding of leg forces. The explanatory part offers a deeper understanding of leg forces concerning performance (maximal velocity) using linear effect models.

### 3.1. Descriptive Results

The average values of the leg force characteristics are presented in Table 2. Interestingly, the velocity increased with each bout together with the stroke rate and different leg force characteristics except the impulse over 10 s, where only the maximal effort displayed a greater impulse than at 12 and 15 km/h. Additionally, it can be seen that the participants’ on average stably maintained the two fixed velocities. 

The leg force profile in Figure 3 reveals that the force elicited a sinus-like pattern, which increased and decreased throughout the stroke cycle. This outcome makes logical sense, according to the cyclic leg movement of paddlers. Additionally, the force peak increased with velocity (Figure 3).

### 3.2. Explanatory Results

Correlations between the maximal velocity and chosen variables, which were the mean stroke rate (SPM), peak force (N), mean force (N), impulse (N⋅s) over one stroke cycle, impulse (N⋅s) over 10 s, and the groups, are presented in Figure 4. The strongest correlation was observed between the maximum velocity and force impulse 10 s (r = 0.827, *p* < 0.01). Multicollinearity could exist between the mean force, peak force, and the two force impulses. The strongest co-correlation was observed between the impulse per stroke cycle and mean force (r = 0.910, *p* < 0.01), which could introduce multicollinearity in the predictive model if present in the same regression.

Table 3 lists the explanatory model results, which used the leg pushing force characteristics as the response variable and velocity as the explanatory variable. Both the univariate and mutually adjusted models exhibited a significant positive relationship with increased velocity and pushing force characteristics (Table 3), which indicates that an increase in paddling velocity is, among other drivers, driven by increased leg pushing force. The R-squared values ranged from 0.14 to 0.34, indicating that leg forces are only part of the mechanism leading to higher paddling velocities. From the table it is evident that the mean pushing force, peak pushing force, and the two impulse variables increased with the velocity. 

The following performance prediction equation was generated from the linear model regression analysis:Vmax = (0.0045 × Fpeak) + (0.0017 × J10 s) + (0.05 × SR) + 7.62.(1)
(F(3,21 = 37.41; *p*-value < 0.001), adjusted R-squared = 0.82)
where Vmax is maximal velocity, Fpeak is the peak force, J10 s is the impulse over 10 s, and SR is the stroke rate. The analysis of variance for this model was significant. Moreover, all predictors displayed a slope that was significantly different from zero (see Equation (1)). This model can predict the maximal kayak velocity using all three predictors with the identified slopes. Using five-fold cross-validation, we identified the predictive model performance as one standard deviation from the observed maximal velocity of each paddler, indicating that the model can correctly predict 68% of the paddlers’ velocity within 1 km/h.

## 4. Discussion

This study is the first to quantify and correlate the paddler’s leg forces generated during kayaking on water in relation to kayak velocity and the results revealed increases in leg forces with velocity. The leg pushing force impulse over 10 s was the best predictor for maximal velocity. Furthermore, all four characteristics of leg pushing force demonstrated a significant positive relationship with an increase in velocity indicating that leg pushing force is a vital factor for increasing kayak velocity.

### 4.1. Descriptive Findings

The mean leg pushing forces found in the current study were on average 77 N higher than the mean leg forces found by Nilsson and Rosdahl [13], while the peak pushing forces were about the were similar in both studies. The five male paddlers in the study by Nilsson and Rosdahl [13] were on average four years older than the male senior paddlers in the current study and included a senior world championship medalist. Since the present study did not have paddlers of that level (Table 1), the forces reported by Nilsson and Rosdahl [13] are expected to be higher. Furthermore, the results in the current study were averaged over 15 stroke cycles, which included the acceleration phase, while Nilsson and Rosdahl [13] averaged the results over 10 s at top velocity; thus, the force might be lower due to the constant velocity and lower drag to overcome. Furthermore, it should be noted that the participants were able to reach and maintain the two fixed velocities. Fluctuation in the fixed velocities could have influenced the results negatively. 

Interestingly, the leg pushing forces in the current study demonstrate similarities to the mean pushing forces found by Begon et al. [27] on the footrest on a kayak ergometer. Minor differences were observed in the mean pushing forces between the two studies. However, the peak pushing forces found by Begon et al. [27] were on average 467 N greater than those in the current study, which could mean that differences exists between footrest peak pushing forces on the kayak ergometer and kayaking on water.

### 4.2. Explanatory Findings

#### 4.2.1. Relationship between Maximal Velocity and Leg Forces

Nilsson and Rosdahl [13] demonstrated that a reduction in the footrest pull and push forces resulted in a significant lowering of kayak velocity. The current study confirms this relationship between leg pushing forces and maximal velocity (Figure 3 and Table 3) since all four leg pushing force characteristics are related to maximal kayak velocity. The regression equation suggests the inclusion of the mean and peak force, force impulse over 10 s, and the stroke rate. The stroke rate has previously been shown to be closely correlated to velocity [10,19] and the current study confirms this relationship.

#### 4.2.2. Correlation between Different Velocities and Leg Pushing Forces

The current study demonstrates that the leg pushing force variables are related to the general velocity also (Figure 4). All predictors displayed a slope that increased with an increase in velocity (Table 3). This result makes sense because the footrest is one of the transfer points for the forces produced on the paddle. The leg extension that produces the pushing force on the footrest induces a pelvic rotation on the seat, enabling an increased stroke length due to the upper body’s forward rotation. Furthermore, including the trunk muscles actively in the paddle’s dynamic movement increases the total muscle mass used to propel the kayak forwards [13,20]. Nilsson and Rosdahl [13] also established a positive relationship between leg forces and velocity. However, in the current study, R-squared values ranged from 0.14 to 0.34, indicating that the leg pushing forces may be only a part of the mechanism leading to higher paddling velocities.

Several authors have suggested that the maintenance of force on the paddle near the force peak throughout the water phase is of great importance [5,10], which would also entail the maintenance of the force on the footrest. A paddler cannot maintain a high force on the paddle without pushing with the stroke leg. Otherwise, the paddle force does not translate to kayak acceleration. An explanation for the correlation between force impulse and velocity, therefore, could be the ability to maintain pushing force in the footrest to entail an efficient stroke. 

#### 4.2.3. Predictive Model

The predictive model predicted 68% of the paddling velocities within 1 km/h (Equation (1)). Several improvements could be made to reduce this error. The residuals versus the fitted plot demonstrated skewness for low and high paddling velocities, indicating much of this prediction error was induced from paddlers not being represented appropriately in this model. This skewness is probably due to the relatively small sample size, as the bootstrap resampling indicated that the data approximated a normal distribution, which could indicate that a larger sample could reduce the prediction error. Further, a larger sample would theoretically reduce the multicollinearity problem, as a larger sample often solves multicollinearity [23]. Another method of reducing the prediction error could be to use the seat or paddle forces with the leg pushing and pulling forces to predict the maximal paddling velocity.

### 4.3. Limitations

The results of the present study only depict part of the overall force picture in sprint kayak, as the seat forces and leg-pull forces also play a role in the stroke. It would have been beneficial to assess these forces as well. Future studies should correlate force data from the footrest, foot strap, seat, and paddle with velocity. In particular, leg-pull forces from the footstretcher could be interesting to investigate, as these forces work in the opposite direction of the movement. The cyclic leg-pull movement is a key part of the stroke as it allows for a greater forward reach due to increased rotation. Nilsson and Rosdahl [13] have shown that the leg-pull forces also are related to velocity. 

Strain gauge instrumentation could be utilized to investigate forces in seat, paddle, and footrest. Several studies have instrumented a paddle with strain gauges [9,14], while no studies have currently implemented strain gauges in the material of the footrest and seat. Therefore, the next logical step could be to instrument the footrest and seat with strain gauges. 

It should be noted that the study by Bonaiuto et al. [28] investigated forces in the paddle and footrest while recording the kayak movement and acceleration with a special developed system called the “e-kayak system”. However, the system was still missing the seat forces in order to elucidate the full kinetic picture. 

## 5. Conclusions

The current study showed that sprint kayakers display changes in the leg extension force profile, which exhibit a positive relationship with velocity. This applies for female and males of different age groups. The study outcome provides a better understanding of the importance of the leg extension force in kayaking. 

Based on the current findings, coaches are advised to ensure that their athletes enhance leg extension actively during the kayak stroke. We propose that the optimal leg extension should result in a large impulse and a high peak force; thus, athletes must push hard with the leg and maintain the force on the footrest during the stroke. This recommendation may be integrated as a training tool to monitor the intensity and velocity, which could be useful for performance development in kayaking.

## Figures and Tables

**Figure 1 sensors-21-06790-f001:**
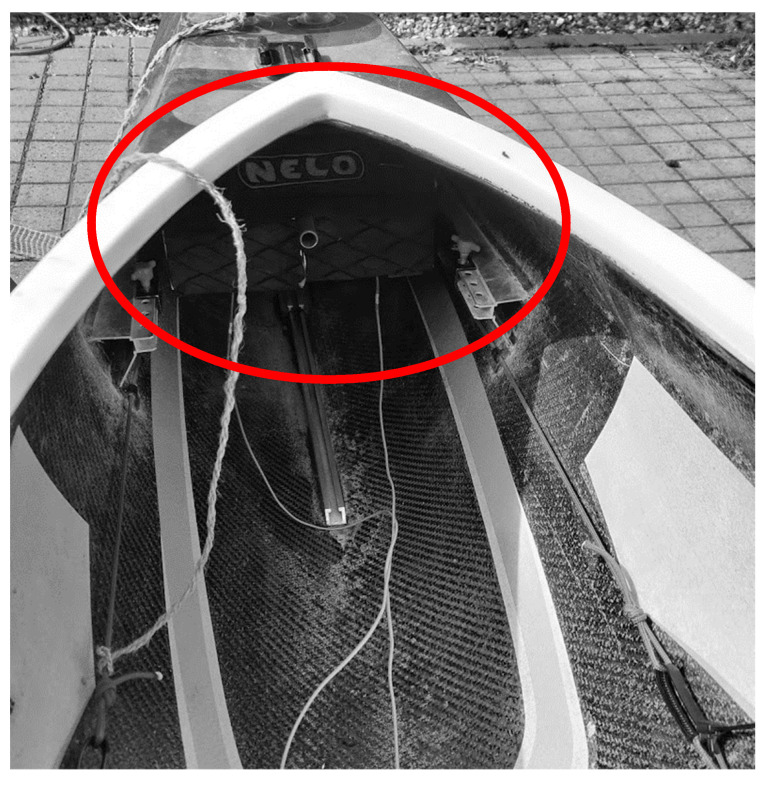
A view of the footrest mounted in a Nelo Cinco kayak. The custom-made footrest fits well in the kayak. It can be seen in the red circle.

**Figure 2 sensors-21-06790-f002:**
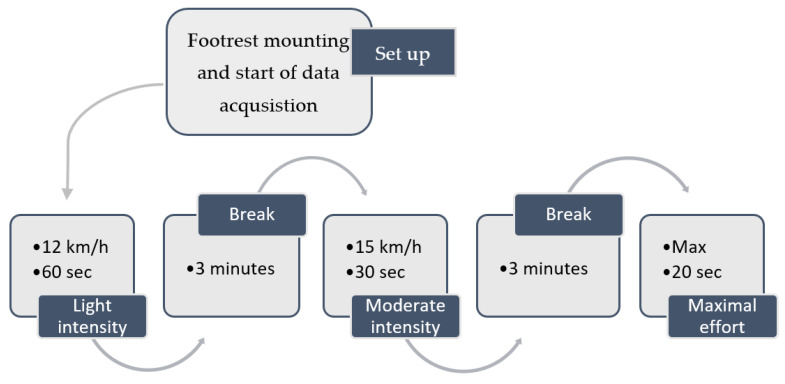
An overview of the workflow of data acquisition.

**Figure 3 sensors-21-06790-f003:**
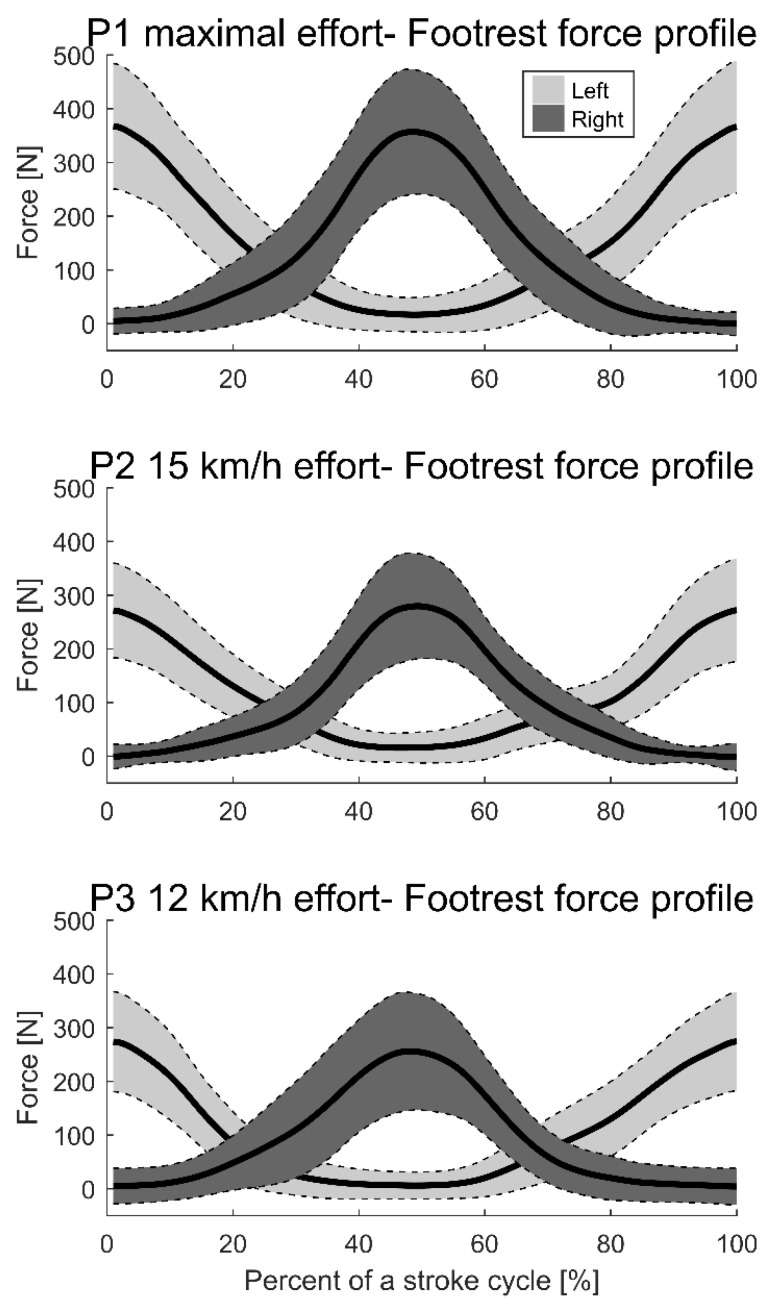
Mean force of stroke cycles for each velocity with standard deviation. Light grey shows the left; dark grey shows the right. P1 is the maximal effort, P2 is 15 km/h, and P3 is 12 km/h. The x-axis presents the stroke cycle from the left side until the return to the left side, and the y-axis shows the force in Newtons. Data are normalized to the left side using the force on the left footrest. The start of the stroke cycle at 0% is when the force on the left footrest peaks. The end of the second stroke at 100% is when the force on the left footrest peaks for the second time.

**Figure 4 sensors-21-06790-f004:**
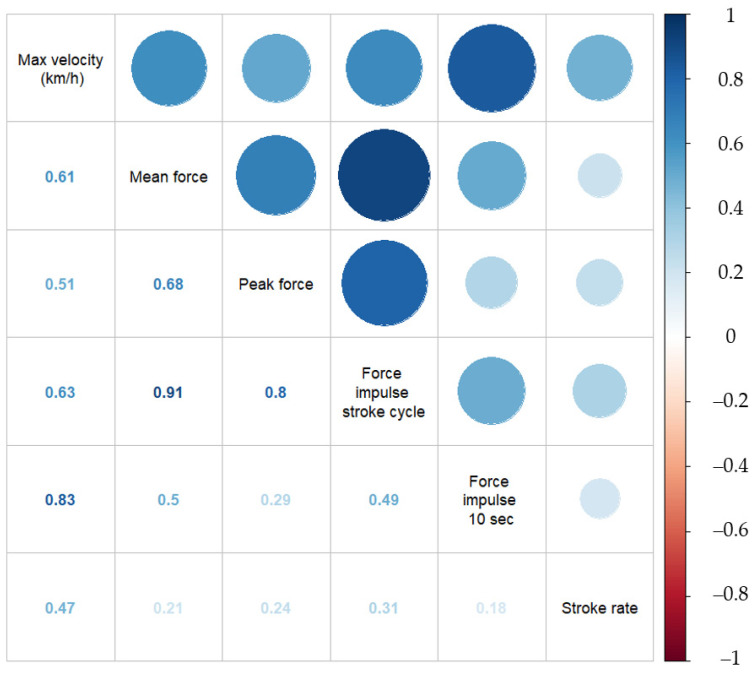
Correlation matrix with correlation coefficients between the maximum velocity and leg force variables (mean force, peak force, force impulse during one stroke cycle, force impulse over 10 s, stroke rate, and group) for 25 paddlers. The coloring indicates different levels of correlations, with dark blue indicating a strong positive correlation, white denoting no correlation, and dark red a strong negative correlation.

**Table 1 sensors-21-06790-t001:** Participant characteristics (*n* = 28).

	Girls under 16 (*n* = 4)	Boys under 18 (*n* = 8)	Senior Women (*n* = 8)	Senior Men (*n* = 8)
Age (years)	14.2 ± 1.2	16 ± 0.8	18.8 ± 1.7	20 ± 1
Weight (kg)	56.7 ± 8.2	73.1 ± 7	63.4 ± 3.9	80.6 ± 7.3
Height (cm)	165.5 ± 6.1	181.7 ± 4.4	167.9 ± 3.2	181.2 ± 6.8
200 m time (s)	52.5 ± 2.3	39.5 ± 2.1	44 ± 1.2	38.8 ± 1.4
500 m time (s)	142.2 ± 6.1	120 ± 4.4	133.45 ± 3.2	112.5 ± 6.8

**Table 2 sensors-21-06790-t002:** The variables averaged for 25 participants.

Biomechanical Variables	12 km/h	15 km/h	Maximal Effort
Velocity (km/h)	12.3 ± 0.5	15.1 ± 0.3	19.7 ± 2
Mean stroke rate (SPM)	74.6 ± 10.6	93.2 ± 11.7	125.0 ± 12
Mean peak force (*N*)	328.0 ± 108.7	347.3 ± 91.6	398.2 ± 106.2
Mean force (*N*)	201.3 ± 67.7	217.7 ± 78	289.5 ± 82.7
Impulse (*N*⋅s) over 1 stroke cycle	397.2 ± 16.5	467.4 ± 15.6	554.8 ± 17.4
Impulse (*N*⋅s) over 10 s	1812.2 ± 72.7	1756.8 ± 69.8	2383.6 ± 86.9

Notes: Two participants withdrew from the test for personal reasons, and data from one participant were lost due to a systematic error. The stroke rate was measured in strokes per minute (SPM).

**Table 3 sensors-21-06790-t003:** Explanatory model.

Biomechanical Variables	Univariate Model (95% CI)	Mutually Adjusted Model (95% CI)
Mean force (*N*) *	14.13 (9.15; 19.08)	13.25 (8.38; 18.12)
Peak force (*N*) *	11.93 (5.15; 18.71)	9.69 (3.31; 16.08)
Impulse 1 stroke cycle (*N*⋅s) *	2.540 (1.471; 3.608)	2.421 (1.342; 3.500)
Impulse 10 s (*N*⋅s) *	110.889 (60.849; 160.929)	104.216 (24.072; 152.226)

Notes: * *p* < 0.05, significant relationship to velocity. The explanatory model was mutual adjusted for the kayak level to elaborate on different potential associations between paddler levels. The slopes and 95% confidence intervals (CIs) of the estimate from the multiple regression analyses to predict maximal velocity are presented.

## Data Availability

The data presented in this study are available on request from the corresponding author. The data are not publicly available due to privacy reasons and regulations.

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
