# Peer review of "Characterization of Leg Push Forces and Their Relationship to Velocity in On-Water Sprint Kayaking"

_sensors, 2021, doi:10.3390/s21206790_

Round 1

Reviewer 1 Report

Dear Authors, this is an important contribution to the area of work in performance technique of sprint kayaking.

My main comment is to review the paper by Kong, P. W., Tay, C. S., & Pan, J. W. (2020). Application of Instrumented Paddles in Measuring On-Water Kinetics of Front and Back Paddlers in K2 Sprint Kayaking Crews of Various Ability Levels. Sensors20(21), 6317 to further improve the literature review.

My secondary comment is regarding further description of the experiment protocol. For example, how was it verified that participants performed at the required speeds of 12 and 15 km/h, what was the rest intervals and was it a single trial? The authors should expect that others may like to replicate the experiment protocol or make comparisons to the results. Hence it is important to have the protocol clearly described in detail.

My other comments as follows:

Typo in Table 1 title. Suggest to split Table 1 to show the 4 sub groups of participants. Not meaningful to combine participant characteristics of U16 girls with Senior Men.

Line 163 - State the version of R-studio.

Table 2 - Typo in column header. For consistency, mean peak force for 12 km/h to have 1 dp. 

Line 294 - grammar.

Author Response

Reviewer 1

Dear Authors, this is an important contribution to the area of work in performance technique of sprint kayaking.

Dear Reviewer 1

We would like to thank you for the response to our manuscript and the opportunity to re-submit and address the comments. Thank you for taking the time to read and comment on our manuscript. The comments were clearly the result of a thorough and very insightful review. We feel that the revisions have improved the manuscript. 

Overall comment

My main comment is to review the paper by Kong, P. W., Tay, C. S., & Pan, J. W. (2020). Application of Instrumented Paddles in Measuring On-Water Kinetics of Front and Back Paddlers in K2 Sprint Kayaking Crews of Various Ability Levels. Sensors20(21), 6317 to further improve the literature review.

Author Response: Thanks for the comment, we have included Kong et al. 2020 in the paper.

Author change to manuscript: A study by Kong et al. (11) used the commercially available paddle system “One Giant Leap” to investigate paddle force profiles of the front and back paddlers in a large sample of K2 sprint kayaking athletes. Kong and colleagues found for the national paddlers, paddle peak force of 344 N and 342 N for the front and back paddlers, and paddle mean force of 205 N and 200 N for the front and back paddlers. These forces seem to be a bit higher than the values obtained by Gomes et al. (10). However, it is essential to note that the study by Kong et al. (11) used crew boats, where the total mass is higher than in K1. Additionally, Kong and colleagues concluded that with instrumented paddles it is possible to obtain performance feedback during on-water kayaking. (Page 2; Line 53:61)

My secondary comment is regarding further description of the experiment protocol. For example, how was it verified that participants performed at the required speeds of 12 and 15 km/h, what was the rest intervals and was it a single trial? The authors should expect that others may like to replicate the experiment protocol or make comparisons to the results. Hence it is important to have the protocol clearly described in detail.

Author Response: Thank you for the comment. The velocity data was checked after data collection, and the average values from the velocity data can be seen in table 2. We have added a sentence to clarify this. We have also created a figure of the workflow of data acquisition where the rest intervals are stated.

Author change to manuscript: Additionally, it can be seen that the subjects were on average good to maintain the two fixed velocities. (Page 7; Line 196-197)

Author change to manuscript: Furthermore, it should be noted that the participants were able to hit and maintain the two fixed velocities. Fluctuation in the fixed velocities could have influenced the results negatively. (Page 11; Line 266-268)

Author change to manuscript: See figure 2 for an overview of the workflow of data acquisition. (Page 5)

Specific comments

My other comments as follows:

Typo in Table 1 title. Suggest to split Table 1 to show the 4 sub groups of participants. Not meaningful to combine participant characteristics of U16 girls with Senior Men.

Author Response: The title in table has been fixed and we have split the subjects into the four groups.

Author change to manuscript: See Table 1(Page 4)

Line 163 - State the version of R-studio.

Author Response: We have stated the version of R-studio

Author change to manuscript: The statistical analyses were conducted in R-studio (v. 1.4.1717), except for bootstrapping, which was conducted in Stata (v. 15; Stata Corp LP, College Station, TX)(Page 6; Line182-183)

Table 2 - Typo in column header. For consistency, mean peak force for 12 km/h to have 1 dp. 

Author Response: The typo has been fixed and the missing decimals has been added

Line 294 - grammar.

Author Response: The grammatical error has been fixed. Note the conclusion has been moved into the discussion due to comments made by reviewer 2. 

Author change to manuscript: Sprint kayakers display changes in the leg extension force profile, which exhibit a positive relationship with velocity. This applies for female and males of different age groups. (Page 12; Line 319-324)

Reviewer 2 Report

The paper describes the use of ad-hoc developed footrest to measure leg push forces and their relationship to velocity in on-water sprint kayaking. 

The reserach work is intereseting but the paper should be better structured. The authors should add a figure to present the workflow of data acquisition. Futher specific pictures should be added to present how the footrest is mounted. 

A more specific descrition must be added about how the footrest works . It is not enough to cite a previous reserach work to present the specific device.

The described limitation should be followed by some solutions available in literature.

The conclusions are not enough for a publication on a scientific journal. The authors should better discuss the reached results and present potential improvements according to the described limitations.

The paper can be accepted after a major revision.

Author Response

Reviewer 2

The paper describes the use of ad-hoc developed footrest to measure leg push forces and their relationship to velocity in on-water sprint kayaking. 

Dear Reviewer 2

We would like to thank you for the response to our manuscript and the opportunity to re-submit and address the comments. Thank you for taking the time to read and comment on our manuscript. The comments were clearly the result of a thorough and very insightful review. We feel that the revisions have improved the manuscript. 

Overall comment

The reserach work is intereseting but the paper should be better structured. The authors should add a figure to present the workflow of data acquisition. Futher specific pictures should be added to present how the footrest is mounted. 

Author Response: Thank you for the comment. We have included a figure of the workflow of data acquisition and included pictures of the mounted footrest. 

Author change to manuscript: See Figure 1 and 2 (Page 4 and 5)

A more specific descrition must be added about how the footrest works. It is not enough to cite a previous reserach work to present the specific device.

Author Response: Thank you for the comment. We have added a detailed description of how the footrest works.

Author change to manuscript: The footrest has the same design as a Nelo footrest. Therefore, it can be fitted to all Nelo kayaks, as this is the most dominant brand on the market. The footrest consists of a metal sandwich with two load cells in between, measuring forces from the left and right leg. The footrest can be seen mounted in the kayak in figure 1.(Page 4; Line 110-114)

The described limitation should be followed by some solutions available in literature.

Author Response: We have added some solutions on how to proceed with this work in the limitations section.

Author change to manuscript:  The results of the present study only depict part of the overall force picture, as the seat forces and leg extension pull forces also play a role in the stroke. It would have been beneficial to assess these forces as well. Future studies should correlate force data from the footrest, foot strap, seat, and paddle with the velocity. Especially leg-pull forces from the foot stretcher could be interesting to investigate, as these forces work in the opposite direction of the movement. The cyclic leg-pull movement are a key part of the stroke as it allows for a greater forward reach due to increased rotation. Nillson and Rosdahl (13) has shown that the leg-pull forces are also related to velocity.

Strain gauge instrumentation could be utilized to investigate forces in seat, paddle and footrest. Several studies have instrumented a paddle with strain gauges (9, 14), while no studies have currently implemented strain gauges in the material of the footrest and seat.  Therefore, the next logical step could be to instrument the footrest and seat with strain gauges. The study by Bonaiuto et al. (28) investigated forces in the paddle and footrest while recording the kayak movement and acceleration with a special developed system called the “e-kayak system”. However, the system is still missing the seat forces in order to elucidate the full kinetic picture.(Page 12; Line 323-340)

The conclusions are not enough for a publication on a scientific journal. The authors should better discuss the reached results and present potential improvements according to the described limitations.

The paper can be accepted after a major revision.

Author Response: Thank you for the comment. We have extended the discussion by presenting potential improvements according to the described limitations, which has been done in one of your previous comments. The conclusion has been incorporated in the discussion while the “practical applications” now stands alone as a single section.

Author change to manuscript:  The current study sprint kayakers display changes in the leg extension force profile, which exhibit a positive relationship with velocity. This applies for female and males of different age groups. The study outcome provides a better understanding of the importance of the leg extension force in kayaking.(Page 12; Line 297-302)

Round 2

Reviewer 2 Report

The authors followed the instructions to improve the quality of the paper. However, the conclusion have been totally removed and changed with a section entitled "Practical applications". A scientific paper should have a final section named "Conclusion".

The paper may be accepted after a minor review.

Author Response

Reviewer 2

Dear Reviewer 2

We would like to thank you for the response to our manuscript and the opportunity to re-submit and address the comments. Thank you for taking the time to read and comment on our manuscript. 

Overall comment

The authors followed the instructions to improve the quality of the paper. However, the conclusion have been totally removed and changed with a section entitled "Practical applications". A scientific paper should have a final section named "Conclusion".

Author Response: Thank you for the comment. We have renamed the section “practical applications” to “conclusion” and moved the conclusion from the discussion and to the new conclusion section.

Author change to manuscript:  The current study sprint kayakers display changes in the leg extension force profile, which exhibit a positive relationship with velocity. This applies for female and males of different age groups. The study outcome provides a better understanding of the importance of the leg extension force in kayaking. Based on the current findings the coaches are advised to ensure that their athletes enhance the leg extension actively during the kayak stroke. We propose that the optimal leg extension should result in a large impulse and a high peak force; thus, athletes must push hard with the leg and maintain the force on the footrest during the stroke. This recommendation may be integrated as a training tool to monitor the intensity and velocity, which could be useful for performance development in kayaking.(Page 12-13; Line 340-350)